# Measurement of Exhaled Nitric Oxide in 456 Lung Cancer Patients Using a Ringdown FENO Analyzer

**DOI:** 10.3390/metabo11060352

**Published:** 2021-05-31

**Authors:** Jing Li, Qingyuan Li, Xin Wei, Qing Chen, Meixiu Sun, Yingxin Li

**Affiliations:** 1Institute of Biomedical Engineering, Chinese Academy of Medical Science & Peking Union Medical College, Tianjin 100730, China; ljinga1118@163.com (J.L.); liqingyuua@163.com (Q.L.); 18639211392@163.com (X.W.); yingxinli@126.com (Y.L.); 2National Clinical Research Center for Cancer, Deparment of Cardio-Pulmonary Function, Cancer Institute and Hospital, Tianjin Medical University, Tianjin 300070, China; chopinliu2006@126.com

**Keywords:** exhaled NO, lung cancer, breath biomarkers, cavity ringdown spectroscopy (CRDS)

## Abstract

The objective of this study was to investigate the clinical value of exhaled nitric oxide (NO) for diagnosing lung cancer patients by using a relatively large sample. An online and near-real-time ringdown exhaled NO analyzer calibrated by an electrochemical sensor at clinical was used for breath analysis. A total of 740 breath samples from 284 healthy control subjects (H) and 456 lung cancer patients (LC) were collected. The recorded data included exhaled NO, medications taken within the last half month, demographics, fasting status and smoking status. The LC had a significantly higher level of exhaled NO than the H (H: 21.0 ± 12.1 ppb vs. LC: 34.1 ± 17.2 ppb). The area under the receiver operating characteristic curve for exhaled NO predicting LC and H was 0.728 (sensitivity was 0.798; specificity was 0.55). There was no significant difference in exhaled NO level between groups divided by different types of LC, tumor node metastasis (TNM) stage, sex, smoking status, age, body mass index (BMI) or fasting status. Exhaled NO level alone is not a useful clinical tool for identifying lung cancer, but it should be considered when developing a diagnosis model of lung cancer by using breath analysis.

## 1. Introduction

Lung cancer is the largest single cause of deaths from cancer in the world, causes approximately 1.8 million deaths worldwide, which is more than the total number of deaths from breast, colon and prostate cancer [1,2]. Screening was introduced with the goal of early detection. The National Lung Screening Trial of America found a lung cancer mortality benefit of 20% and a 6.7% decrease in all-cause mortality with the use of low-dose chest computed tomography (LDCT) in high-risk individuals [3]. However, there is still a need to evaluate risks associated with radiation exposure and the relatively high rate of false-positive results. To date, histological biopsy remains the gold standard diagnostic method in cancer even though it is invasive, risky, time-consuming and expensive [4]. Therefore, there is an urgent need for inexpensive and noninvasive diagnostics to promote the early detection of lung cancer.

Breath analysis is a strong candidate for lung cancer detection since it is noninvasive, easy to use, low-cost and can detect very low concentrations of volatile components. An increasing number of studies have screened volatile organic compounds (VOCs) as breath biomarkers to diagnose lung cancer. However, the lack of reproducibility of breath biomarkers between different studies restricts further clinic applications of these biomarkers. Recently, in addition to the research on VOCs as breath biomarkers for the diagnosis of lung cancer, some studies have concluded that the level of exhaled nitric oxide (NO), one of inorganic constituents, in lung cancer patients (LC) is higher than that in healthy control subjects (H) [5,6,7], and mounting evidence has indicated that NO signaling is implicated in the pathophysiology of lung cancer [8,9]. Many factors can lead to the increase of exhaled NO level in lung cancer patients, such as T-helper cell 2, nitric oxide synthase, dietary habits, etc. [5]. Measurement of exhaled NO through the use of external devices is a non-invasive and simple technique regarded as a potentially valuable tool for the screening and follow-up of lung cancer [10]. However, evidence for the diagnosis of lung cancer by exhaled NO is not sufficient, and previous studies on measuring exhaled NO were based on small pilot studies (the sample size ranged from 11 to 164). Due to the heterogeneity and variety of physiological and pathological backgrounds of patients, the data on breath analysis are of less statistical significance if sampling is insufficient. Therefore, large-scale breath analysis was carried out in this study to evaluate whether exhaled NO is a reliable breath biomarker for diagnosing lung cancer.

At present, there are several technologies available for detecting exhaled NO that are based on chemiluminescence, electrochemical sensing and laser-based detection [11]. Chemiluminescence instruments can be very sensitive, with detection limits at ppb level of 1.5 (for NIOX from Aerocrine, Solna, Sweden) or even lower (0.1 ppb for CLD 88 from Eco Medics, Duernten, Switzerland; 0.5 ppb for Sievers NOA 280i from GE Analytical Instruments). The response time of such a system is fast, between 0.5 and 0.7 s. In addition to the large size and high cost of investment and running, chemiluminescence analyzers need to be calibrated frequently [10,12,13]. The electrochemical sensors used for breath tests have advantages due to their small size, low price and detection limit of 5 ppb, but they still face the problem of poor reproducibility of exhaled NO measurement results [14,15,16]. Several laser-based spectral techniques for detecting FENO have been reported [13,17,18,19,20,21,22,23,24,25,26]. Recently, we constructed a ringdown FENO analytic system based on the cavity ringdown spectroscopy (CRDS) technique to pursue large-scale, online clinical testing in near real time. The CRDS technique has high sensitivity and high selectivity and has been successfully used for trace-gas analysis, including breath-gas analysis [27,28]. CRDS also has the advantages of fast response and relatively low cost, making the technique both scientifically and economically attractive for breath analysis, particularly when a large number of subjects have to be tested.

In this study, the exhaled NO levels of 740 breath gas samples (including 284 healthy control samples, and 456 primary lung cancer samples) were measured by ringdown exhaled NO analytic system. The system has the advantages of having a fast response, a high data throughput and validated accuracy. In addition to determine whether the exhaled NO level was elevated in LC, we also analyzed the associations between different determinants (sex, age, smoking, etc.) and exhaled NO. This is the largest number of LC patients to be recruited in a human exhaled-NO breath analysis study to date. The results were used to evaluate the clinical application potential of exhaled NO in diagnosing LC, which will help guide the development of new predictive methods based on breath analysis.

## 2. Results

### 2.1. Patients

A total of 740 breath gas samples were collected from 456 primary LC with pathologically diagnosed lung cancer without prior anticancer treatment (e.g., therapeutic agents, radiotherapy or chemotherapy), and 284 healthy control subjects. The demographics, tumor characteristics, smoking status and fasting status of the LC and H are summarized in Table 1.

The mean age of the 456 primary LC (256 males, 56%) was 60 ± 8 years (age range: 29–81 years), and the common smoking status of the patients was non-smoker, accounting for 45%. The main tumor cell types were adenocarcinoma (73%), squamous cell carcinoma (16%) and small-cell carcinoma (10%). More than 36% of the subjects had stage I cancer.

The mean age of the 284 H (159 males, 60%) was 47 ± 14 years (age range: 22–86 years), and 191 of the subjects were non-smokers (68%).

As shown in Figure 1 (left), exhaled NO has a skewed distribution, so we show the results as the median (25% and 75%) of exhaled NO concentration. The number of H with exhaled NO < 20 ppb was significantly higher than that of LC. The number of H decreased significantly with increasing exhaled NO level, eventually becoming lower than the number of LC. The highest level of exhaled NO in H was below 60 ppb, and the highest in LC was above 100 ppb. As shown in Figure 1 (right), the median exhaled NO level of the 284 H was 19.0 (11.4, 30.1) ppb, whereas the median level of the 456 LC was 32.0 (21.8, 44.8) ppb. The mean level of exhaled NO from LC was higher than that from H (LC: 34.1 ± 17.2 ppb, H: 21.0 ± 12.1 ppb, according to Mann–Whitney *U*-test: *p*-value < 0.01). The area under the curve (AUC) for predicting the lung cancer and healthy control groups was 0.728 (the 95% confidence interval was 0.692 to 0.765), the sensitivity was 0.798 (the 95% confidence interval was 0.758 to 0.834), and the specificity was 0.55 (the 95% confidence interval was 0.490 to 0.608) (Figure 2).

### 2.2. The Level of Exhaled NO among Patients with Different Types of Lung Cancer

In the present study, the most common histology was adenocarcinoma, followed by squamous cell carcinoma and small cell carcinoma. The exhaled NO levels observed in the various types of lung cancer are presented in Table 2. The exhaled NO levels in the adenocarcinoma group (31.5 ppb) was no different from the squamous cell carcinoma group (31.6 ppb, according to Kruskal–Wallis test: *p*-value = 1), but higher than those in the small-cell lung cancer group (24.0 ppb, according to Kruskal–Wallis test: *p*-value = 0.05) (Figure 3). The box and scatter plot in Figure 3 show the range of exhaled NO in patients with various types of lung cancer (adenocarcinoma: 3.2–94.7 ppb; squamous cell carcinoma: 4.6–78.5 ppb; small-cell lung cancer: 7.0–82.1 ppb). There were no statistically significant differences in exhaled NO level between patients with different types of lung cancer.

### 2.3. Comparison of Exhaled NO Level in LC across Different TNM Stages

Lung cancer diagnosis also requires determination of the extent of the tumor to define the TNM stage, which will ultimately guide cancer treatment options. Some scholars believe that tumor pathogenesis is related to excessive or inappropriate production of NO [29,30]. NO in the human body is a kind of active free radical with extensive and multiple effects. NO can exhibit either tumor promotion or antitumor activity: a continuous low concentration of NO can promote tumor growth, whereas a high concentration of NO has an antitumor effect [31,32,33]. Therefore, the exhaled NO levels in different stages of lung cancer are presented in Table 3. The exhaled NO level was highest in stage III LC and lowest in stage 0 (carcinoma in situ) LC (stage 0, 29.4 (17.8, 41.0) ppb; stage I, 31.6 (22.1, 44.0) ppb; stage II, 31.6 (21.9, 41.9) ppb; stage III, 33.6 (22.9, 45.5) ppb; and stage IV, 31.0 (20.5, 48.4) ppb). It can be seen from the scatter and bar chart (Figure 4) that early stage (stage I) lung cancer was significantly more common in the concurrent group, and the level of exhaled NO in each group was as follows: stage 0, 5.9–52.3 ppb; stage I, 3.72–82.4 ppb; stage II, 3.2–70.6 ppb; stage III, 9.2–93.7 ppb; and stage IV, 3.3–107.5 ppb.

The results of exhaled NO with respect to sex, smoking status, age, BMI and fasting status in the lung cancer group and healthy control group are presented in Table 4. Females had no different levels of exhaled NO than males in both LC and H groups (LC-male: 30.9 (21.6, 42.6) ppb, LC-female: 32.2 (22.0, 46.0) ppb, H-male: 19.7 (12.3, 29.7) ppb, H-female: 18.0 (9.6, 31.1) ppb, all *p* > 0.05). The median exhaled NO level of the non-smoker group was higher than that of the smoker and ex-smoker groups. In this study, all of the patients were hospitalized patients, who were not allowed to smoke, so the duration of abstinence in the smoking group ranged from one day to four months. The smoking group of healthy subjects also did not smoke on the day of sampling. This may account for the smaller difference between the smoker group and the non-smoker group. Sex, height and age are also variables that needs to be considered too. The following, subjects were divided into four groups with the age of 50 as the boundary. It can be seen that, in the LC group, the median exhaled NO level of the greater than 50 years old group was greater than that of the less than 50 years old group, while that in H group was opposite. The threshold value of BMI group was 23.9, and the median exhaled NO level of the group with BMI greater than 23.9 was higher than that of the group with BMI less than 23.9. As mentioned above, the differences between healthy groups in all domains were weak. We also performed the same statistical analyses with patients from fasted and fed groups. In the lung cancer group, the exhaled NO levels were almost the same in the fasted and fed groups, whereas in the healthy control group, the exhaled NO level was higher in the fasted group than in the fed group. There were no significant differences between the groups divided by sex, smoking status, age, BMI or fasting status.

## 3. Discussion

Variables such as sex, current asthma, allergic rhinitis, personal history of tobacco use, current use of inhaled corticosteroids, atopy, seasonality and rural versus urban setting have all been previously identified as important explanatory factors that influence exhaled NO levels [34]. In some studies, the concentration of exhaled NO was measured online by chemiluminescent technique at the exhaled flow rate of 250 mL/s. The results showed that the concentration of exhaled NO was higher in those with eosinophilic bronchitis, asthma, atopic or allergic rhinitis than normal controls [35,36,37,38]. In this study, we measured the exhaled NO levels of 740 breath samples by using a CRDS exhaled NO test system that was built in-house and demonstrated that the level of exhaled NO was greater in primary LC than in H. In previous studies, the absolute concentrations of exhaled NO obtained by separate workers in similar patient groups and normal subjects with apparently similar techniques have been very different. In 1998, Liu [6] used chemiluminescent technique to measure the concentration of exhaled NO when the exhaled flow rate was 250 mL/s. The level of exhaled NO from cancer patients (16.9 ± 0.9 ppb; *n* = 28) was significantly higher than from the control group (6.0 ± 0.5 ppb; *n* = 20, *p* < 0.001). In 2018, Liu [5] recruited 172 H and 164 LC with pathologically diagnosed. The concentration of exhaled NO was measured online by a nitric oxide analyzer at the exhaled flow rate of 50 mL/s. The LC had a significantly higher level of exhaled NO than the H (33.9 ± 15.6 ppb, *n* = 163; 16.8 ± 4.2 ppb, *n*  = 172; *p* < 0.01). In 2005, Masri [7] used a chemiluminescent analyzer to measure exhaled NO levels via the off-line method. It was demonstrated that LC had higher levels of exhaled NO compared with control subjects (control subjects, 7.4 ± 0.5, *n* = 35; LC, 18.4 ± 3.2 ppb, *n* = 11; *p* = 0.001). However, within studies, significant differences between LC and H will still be valid.

In fact, interpreting exhaled NO levels in clinical practice is even more complex than in academic studies. Reference values that consider background characteristics such as sex, age and smoking may indeed be useful in guiding the interpretation of exhaled NO values in adults. Some studies have shown that exhaled NO level is decreased by cigarette smoking [39,40]. Taylor [41] reported that the level of exhaled NO is associated with sex, but Olin [42] reported that it is height, not sex, that is associated with exhaled NO level. In 2018, Liu [5] demonstrated that exhaled NO was higher in the squamous cell carcinoma group than in the adenocarcinoma group, and exhaled NO was not significantly different between the small-cell lung cancer group and the other two groups. In this study, adenocarcinoma accounted for 73% of lung cancer cases, and 50% of patients were diagnosed at stage I or II. There was no significant difference in exhaled NO level between patients with different types of lung cancer, which may be due to the uneven proportion of lung cancer types in this study. The next study will be to increase the number of subjects with squamous cell carcinoma and small cell carcinoma, eliminate individual differences and verify the role of exhaled NO in different types of lung cancer. We compared the level of exhaled NO between the groups divided by lung cancer stage and fasting status for the first time. In addition, there were no differences between the groups divided by lung cancer stage, sex, smoking status, age, BMI or fasting status. Our results (the AUC was 0.728, the sensitivity was 0.798 and the specificity was 0.55) suggest that exhaled NO level alone is not sufficient to ascertain a lung cancer diagnosis. Next, we will study the metabolic mechanism of exhaled NO in lung cancer patients and establish a lung cancer diagnosis model by using exhaled NO and VOCs.

## 4. Materials and Methods

### 4.1. Subjects

From November 2018 to October 2019, 456 breath-gas samples were collected from LC (164 smokers, 86 ex-smokers and 206 non-smokers) at different cancer stages who were hospitalized in Tianjin Medical University Cancer Institute and Hospital (average age 60 years with a range of 29–81 years) before lung cancer diagnosis. The clinical status of all LC was confirmed by pathological diagnosis within one month after sampling. Samples of lung tissue lesions were obtained by bronchoscopy or surgery for pathological examination, including typing and staging. None of the patients had symptoms of cough, expectoration or dyspnea, and no asthma, chronic obstructive pulmonary disease or other symptoms were found in the pathological diagnosis. The LC was compared with 284 H (72 smokers, 19 ex-smokers and 191 non-smokers; average age 47 years and age range of 22–86 years). Healthy subjects were people without pulmonary disease and without any inflammation for at least one month, and they were also recruited at Tianjin Cancer Hospital. CT images of the lungs did not show any pulmonary nodules. Informed consent was obtained from all subjects involved in the study.

### 4.2. Ethics Approval

The study was conducted according to the guidelines of the Declaration of Helsinki and approved by the ethics committees of the Tianjin Cancer Hospital. The trial was registered with the Institutional Review Board (IRB) of the Chinese Clinical Trial Registry (registration number: chiCTR1900023659). All methods were carried out in accordance with relevant guidelines and regulations, and informed consent was obtained from all participants.

### 4.3. Sampling of Exhaled NO

Exhaled NO levels relative to breath volume were measured by using a ringdown exhaled NO analytic system based on cavity ringdown spectroscopy (CRDS) and expressed in parts per billion (ppb). Fluorinated ethylene propylene (FEP) bags (1 L, Beijing HCTC Environmental Protection Technology Ltd., Beijing, China) were used to collect off-line breath samples. Previously, we proved that FEP bags can store standard NO gas diluted by nitrogen for 5 h without significant changes [43]. In this study, each subject remained in a sitting position for approximately 3 min and then provided one breath sample (>0.7 L) in a sitting position through a disposable mouthpiece between 10 and 15 s, without using a nose-clip, in accordance with the recommendations from the European Respiratory Society and the American Thoracic Society [44,45]. All samples were collected between 8:00 a.m. and 11:00 a.m., placed in a cooler box (28.4 L, Coleman, Chicago, IL, USA) to avoid light and keep them at a constant temperature, and processed within 5 h. Before collection of breath, all bags were thoroughly cleaned to remove any residual contaminants by flushing with nitrogen gas (purity of 99.9999%) more than three times.

### 4.4. Ringdown Exhaled NO Analyzer

Several laser-based spectral techniques for detecting exhaled NO have been reported [13,17,18,19,20,21,22,23,24,25,26]. The wavelengths used are all mid-infrared near 5.2 μm (ranging from 5.1 to 5.7 μm). In this spectral range, several other gases, such as CO_2_ and H_2_O, also create absorption bands. In addition, according to the HITRAN database, the strongest intermediate infrared absorption cross-section is 3.89 × 10^−19^ cm^2^/molecule. Compared with this vibrational band, the electronic transition of NO in the ultraviolet region near 226 nm has a higher absorption strength and cross-section. The UV region near 226 nm is higher, which corresponds to the electron transition. The absorption cross-section near 226 nm is 1.87 × 10^−18^ cm^2^/molecule, four times higher than that of 5.2 μm [46]. This shows that 226 nm can improve the detection sensitivity of NO. In addition, water does not interfere with the measurement of NO at 226 nm, which indicates that 226 nm is more advantageous than 5.2 μm. Recently, we constructed a ringdown exhaled NO analytic system with 226 nm wavelength as laser source. And we verified the measurement accuracy of this CRDS exhaled NO analyzer by comparing with an electrochemical sensor in General Hospital of Tianjin Medical University. The results showed significant correlation between these two methods, with a concordance correlation coefficient of 0.90 (*p*-value < 0.05, Figure 5, left). As shown in Figure 5 (right), the agreement between the concentration of exhaled NO of the two methods was assessed by using the Bland–Altman analysis, which illustrated good consistency. Comparison of the testing results shows that this ringdown breath acetone analyzer can be used for reliable online, real-time clinic testing. The overall structure, working principle and system performance of the experimental device are described in detail in our previous work [43], and this device has been previously validated and demonstrated to be reliable and advantageous. The system has a lower detection limit of 7.4 ppb, and the baseline stability of the system is 0.52%, with good repeatability, stability and real-time performance. The linear response of the CRDS system (R = 0.988) and the accuracy of the dual wavelength background differential method were verified by using a mixture of human respiratory gas and standard NO gas. This work marks the first time CRDS was used for a lung-cancer study.

#### Measuring Method of Exhaled NO

The background-subtraction method was used to determine the exhaled NO level in breath samples. This method is described in detail in our previous work, the exhaled NO level can be determined from a difference in the absorbance between a selected peak and the background baseline [43]. The wavelength of 226.255 nm was chosen as the most suitable wavelength to detect exhaled NO because many VOCs in the breath, such as acetone and isoprene, absorb radiation with a wavelength of 226 nm. Moreover, NO shows no absorbance at 226.290 nm, so the absorbance of breath gas at the wavelength of 226.290 nm was used as the background baseline. Then the absolute concentration of exhaled NO in the breath gas was obtained by Equation (1):(1)$$A_{\left(NO\right)}=A_{\left(226.255\mathrm{nm}\right)}-A_{\left(226.290\mathrm{nm}\right)}=n\sigma_{\left(226.255\mathrm{nm}\right)}d=\frac{d}{c}\left(\frac{1}{\tau_{226.255}}-\frac{1}{\tau_{226.290}}\right)$$
where *A*_(*NO*)_ is the absorbance of exhaled *NO*; *A*_(226.255nm)_ is the absorbance of breath gas at 226.255 nm; *A*_(226.290nm)_ is the absorbance of the baseline of the breath sample; *σ*_(226.255)_ is the absorbance cross-section of *NO* at 266.255 nm, which was determined to be 7.64 × 10^−18^ cm^2^ molecule^−1^ at room temperature and 760 Torr; n is the sample concentration; d is the distance between the two mirrors, which was 47 cm; *c* is the speed of light; and $\tau_{226.255}$ and $\tau_{226.290}$ are the ringdown times of breath gas detected at 226.255 and 226.290 nm, respectively.

### 4.5. Self-Reported Outcomes

Information about lung disease history, medicine use, fasting and tobacco smoking was obtained through self-reporting. The history of lung disease was defined as an affirmative response to the following question: “Have you ever had lung disease?” Use of medicines was defined as subjects having taken any kind of medicine (including sprays, pills, capsules and decoctions) within half a month. Fed was defined as an affirmative answer to the following question: “Have you eaten breakfast already?” The smoking statuses were non-smoker, ex-smoker and smoker. The ex-smoker status was defined as having quit smoking four or more months before the study. For smokers and ex-smokers, the amount of smoking (number of cigarettes per day) was determined.

### 4.6. Statistical Analysis

All statistical tests were two-sided, and a *p*-value of < 0.05 was adopted for statistical significance. All analyses were performed by using the Statistical Product and Service Solutions (IBM SPSS Statistics) version 25.0 (IBM Inc., Endicott, NY, USA). Standard formulas were used for the analysis. The data did not approximate a normal distribution, so nonparametric statistical analyses were used. Differences between two groups were determined by the Mann–Whitney *U*-test (two-tailed), and a subsequent analysis was performed by using Kruskal–Wallis tests to assess the significance of differences between groups.

## 5. Conclusions

In conclusion, due to the high data throughput of the near-real-time online ringdown exhaled NO analyzer, this study was able to analyze a significantly large number of subjects to provide new data regarding exhaled NO levels in LC. The results show that the levels of exhaled NO were significantly increased in LC compared with H. However, our results suggest that exhaled NO level alone is not sufficient to ascertain a lung cancer diagnosis. Additional studies on the use of exhaled NO combined with other volatile organic compounds (VOCs) to develop a diagnosis model of lung cancer are needed.

## Figures and Tables

**Figure 1 metabolites-11-00352-f001:**
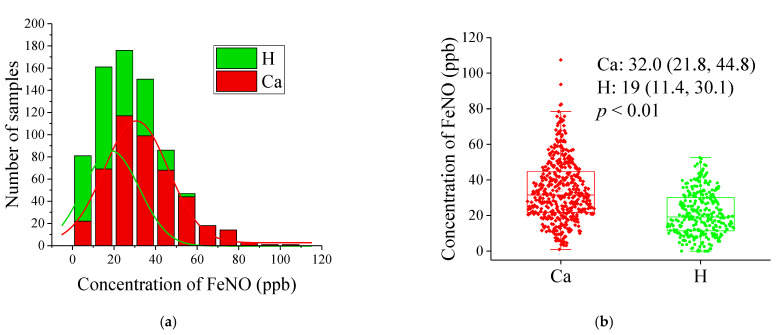
(**a**) The number of samples in each 10 ppb interval of exhaled NO concentration for both H and LC. (**b**) exhaled NO levels in the LC (red) and H (green). The mean exhaled NO was significantly higher in the LC than in the H. The results are presented as medians (25% and 75%).

**Figure 2 metabolites-11-00352-f002:**
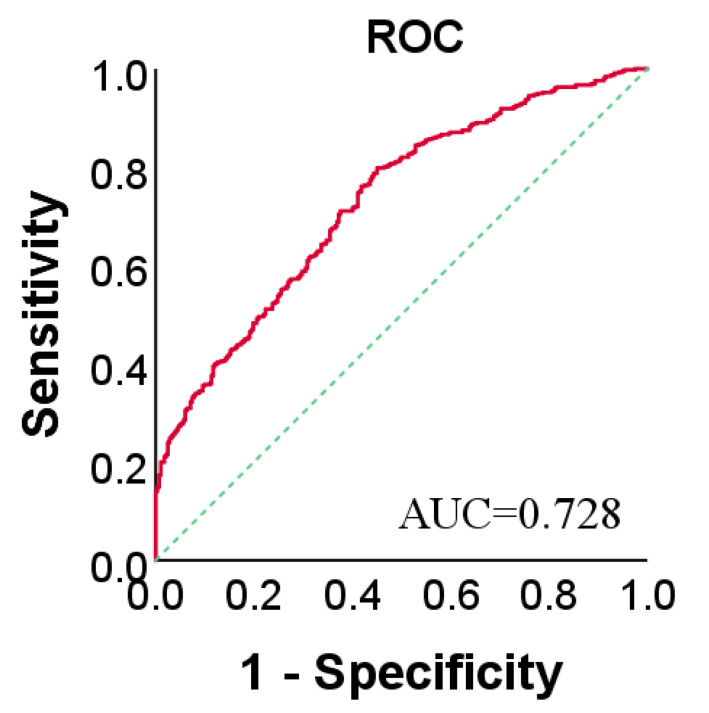
Receiver operating characteristic (ROC) curves for exhaled NO in LC (*n* = 456) and H (*n* = 284).

**Figure 3 metabolites-11-00352-f003:**
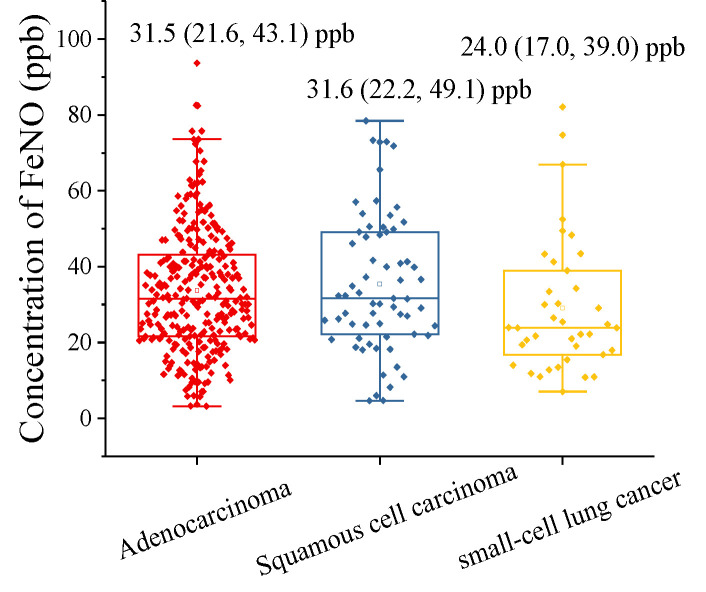
The exhaled NO levels in the squamous cell carcinoma, adenocarcinoma and small-cell lung cancer groups. exhaled NO levels presented as medians (25% and 75%).

**Figure 4 metabolites-11-00352-f004:**
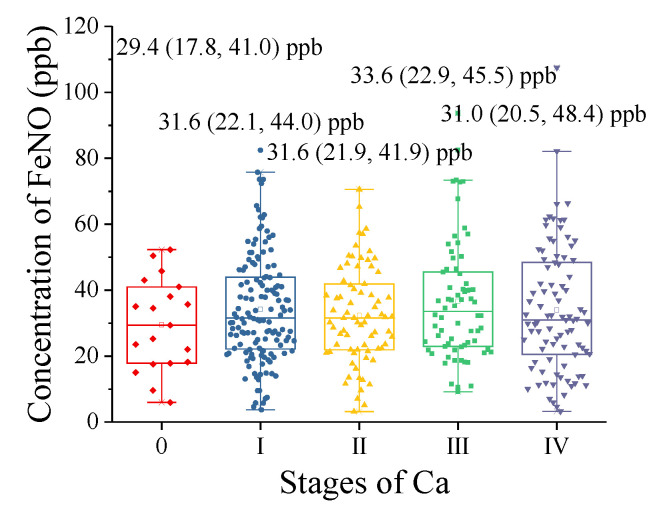
The exhaled NO levels in LC across different TNM stages (stage 0: carcinoma in situ). exhaled NO levels presented as medians (25% and 75%).

**Figure 5 metabolites-11-00352-f005:**
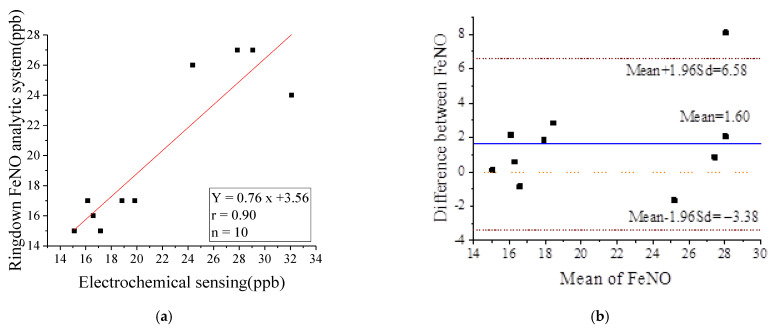
The linear relationship between two methods for measuring exhaled NO (**a**). Bland–Altman analysis of two techniques of measuring exhaled NO concentration (**b**).

**Table 1 metabolites-11-00352-t001:** Baseline information for all subjects.

	Lung Cancer (*n* = 456)	Heathy Control (*n* = 284)
Male (%)	256 (56%)	159 (56%)
Age (range)	60 ± 8 (29–81)	47 ± 14 (22–86)
Smokers	164	72
Ex-smokers	86	19
Non-smokers	206	193
BMI	24.35 ± 3.32	23.88 ± 3.22
Fasting (%)	166 (36%)	253 (89%)
Adenocarcinoma (%)	292 (64%)	NA
Squamous cell carcinoma (%)	65 (14%)	NA
Small-cell lung cancer (%)	40 (9%)	NA
0 (%)	19 (5%)	NA
I (%)	140 (35%)	NA
II (%)	74 (19%)	NA
III (%)	70 (18%)	NA
IV (%)	90 (23%)	NA
EXHALEDNO (ppb)	32.0 (21.8, 44.8)	19 (11.4, 30.1)

Note: 0, LC at stage 0; I, LC at stage I; II, LC at stage II; III, LC at stage III; IV, LC at stage IV.

**Table 2 metabolites-11-00352-t002:** Exhaled NO levels in patients with three different types of lung cancer.

Subtype	*n* (%)	Exhaled NO (Medians (25% and 75%), ppb)	*p*-Value
Adenocarcinoma	292 (74%)	31.5 (21.6, 43.1)	*p* = 0.064 > 0.05
Squamous cell carcinoma	65 (16%)	31.6 (22.2, 49.1)	
Small-cell lung cancer	41 (10%)	24.0 (17.0, 39.0)	

**Table 3 metabolites-11-00352-t003:** Exhaled NO levels in patients with different lung cancer stages.

Stage	*n* (%)	Exhaled NO (Medians (25% and 75%), ppb)	*p*-Value
0	19 (5%)	29.4 (17.8, 41.0)	*p* = 0.685 > 0.05
I	140 (35%)	31.6 (22.1, 44.0)	
II	74 (19%)	31.6 (21.9, 41.9)	
III	70 (18%)	33.6 (22.9, 45.5)	
IV	90 (23%)	31.0 (20.5, 48.4)	

**Table 4 metabolites-11-00352-t004:** The exhaled NO level in sex, smoking status, age, BMI and fasting status groups.

Group		*n*	Exhaled NO (Medians (25% and 75%), ppb)	*p*-Value
Lung cancer	Smoker	164	30.2 (20.6, 41.9)	0.064
	Ex-smoker	86	29.7 (20.8, 40.8)	
	Non-smoker	206	33.8 (23.5, 47.2)	
	Fasted	166	31.5 (21.0, 43.8)	0.811
	Fed	260	31.4 (21.7, 43.0)	
	MALE	256	30.9 (21.6, 42.6)	0.481
	FEMALE	198	32.2 (22.0, 46.0)	
	Age > 50	395	31.9 (21.9, 45.8)	0.104
	Age < 50	58	27.7 (20.6, 38.1)	
	BMI < 23.9	213	31.4 (21.6, 45.5)	0.284
	BMI > 23.9	247	31.9 (22.0, 44.1)	
Healthy control	Smoker	72	26.9 (16.5, 41.7)	1
	Ex-smoker	19	22.5 (20.3, 33.3)	
	Non-smoker	193	30.6 (25.1, 47.3)	
	Fasted	253	19.5 (11.8, 30.0)	0.197
	Fed	29	13.1 (7.8, 31.1)	
	MALE	159	19.7 (12.3, 29.7)	0.59
	FEMALE	123	18.0 (9.6, 31.1)	
	Age > 50	111	18.7 (12.3, 28.0)	0.866
	Age < 50	171	19.4 (10.2, 31.2)	
	BMI < 23.9	146	19.2 (11.6, 30.1)	0.984
	BMI > 23.9	136	19.3 (10.9, 30.0)	

## Data Availability

The data that support the findings of this study are available from the corresponding author upon reasonable request. The data are not publicly available due to the privacy of subjects.

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
