# Peer review of "Measurement of Exhaled Nitric Oxide in 456 Lung Cancer Patients Using a Ringdown FENO Analyzer"

_metabolites, 2021, doi:10.3390/metabo11060352_

Round 1

Reviewer 1 Report

This paper reports on the use of NO concentration testing for the diagnosis for lung cancer. The paper is interesting and the information provided was new. The statistics were appropriate and the plots clear and well presented. The English was poor in places and will need addressing. I have a number of small comments, which are listed below, but my only major concern was the lack of important aspects in the discussion, for example whether the results indicate this is a clinically relevant test? How could it be improved? What is the cost point of the test? Etc.
Line 27: This is a grammatically poor opening line and this and the reference needs fixing
Line 33: What was the false positive rates?
Line 39: There have been numerous papers on lung cancer VOC detection, either directly or through capture and subsequent analysis. I would expect the authors to discuss this.
Line 41: Ca and H need defining in the main text, not just in the abstract
Line 46: How many samples and what was the result of these other trials?
Line 48: I am not sure what point you are trying to make.
Line 52: This paragraph seems overly long and technical for an introduction. I would suggest reducing and putting some in the methods.
Line 80: Poor grammar
Table 1: Were the lung cancer patients receiving any form of treatment or medication? This should be included.
Line 111: Please add confidence interval values for AUC, sensitivity and specificity. I would also like to know (numerically) how many it got correct and what was the false positive rate. In addition, please provide the threshold for the ROC used.
Line 119: I found it odd that the methods nor the introduction discuss the use of electrochemical NO sensors – especially after the comments associated with them in the introduction. I think you should reduce this section and add it to the methods to say how the instrument was calibrated.
Line 143: You should be clear what the p value refers to. Be specific on the comparison.
Line 177: Choosing 50 seems quite arbitrary. Would it not have been better to use the average age of a lung cancer sufferer?
Line 179: Should not start a sentence with “And”.
Lin 206: I would expect this to be linked with the breath analysis collection approach and I would like to see some comparison in the methods used.
Line 221: I would like to know the limitations of the study and further explanation for the results. For example, the analysis on cancer type is not discussed in the discussion. All the points raised in the results should be discussed/commented upon in this section. There is also no discussion of the clinical importance of the results. Is the sensitivity/specificity sufficient to be used as a screening test? What could be improved? These need to be included.
Line 250: Please add details of bag volume and air collected volume. It would be useful for the reader without the need to look at previous work.
Line 265: How was stability measured or is this from the manufacturer?
Line 271: Brief details should be provided in the paper.

Author Response

Q1: Line 27: This is a grammatically poor opening line and this and the reference needs fixing

A1: We are sorry for the mistake. We have corrected “As a malignant tumor with characteristically high morbidity, lung cancer is one of the leading causes of cancer death worldwide. It kills approximately 1.8 million people each year, more than breast, colon, and prostate cancers combined.” to “Lung cancer is the largest single cause of deaths from cancer in the world, causes approximately 1.8 million deaths worldwide, which is more than the total number of deaths from breast, colon, and prostate cancer.”. The reference has been modified according to the Reviewer’s comments in the revision. (Page 2, Lines 26-30)

Q2: Line 33: What was the false positive rates?

A2: False Positive: the truth is negative, but the test predicts a positive. The person is not sick, but the test inaccurately reports that they are. Also called a Type I error in statistics. The false positive rate is calculated as FP/FP+TN, where FP is the number of false positives and TN is the number of true negatives (FP+TN being the total number of negatives). It’s the probability that a false alarm will be raised: that a positive result will be given when the true value is negative. (FP: False Positive, TN: True Negative)

Q3: Line 39: There have been numerous papers on lung cancer VOC detection, either directly or through capture and subsequent analysis. I would expect the authors to discuss this.

A3: Thank you for your suggestion. We discussed the detection of VOCs in lung cancer according to the Reviewer’s comments in the revision. (Page 2, Lines 40-44)

Q4: Line 41: Ca and H need defining in the main text, not just in the abstract.

A4: Thank you for your suggestion. According to the opinions of other reviewers, LC was used instead of lung cancer patients, The LC and H have been modified according to the Reviewer’s comments in the revision. (Page 2, Line 46)

Q5: Line 46: How many samples and what was the result of these other trials?

A5: Thank you for your suggestion. The sample size of lung cancer patients in the references ranged from 11 to 164, which we have added in the revision (Page 3, Lines 53-54). The summary statements are on lines 43 to 46 in the revision, and the specific discussion is in the discussion section, which shows the sample size and results of the references. (Pages 13-14, Lines 216-226)

Q6: Line 48: I am not sure what point you are trying to make.

A6: Sorry, we didn't make it clear. We have corrected “Therefore, large-scale breath analysis with fast response, high data throughput, and validated accuracy was carried out in this study to evaluate whether exhaled FENO is a reliable breath biomarker for diagnosing lung cancer.” to “Therefore, large-scale breath analysis was carried out in this study to evaluate whether exhaled NO is a reliable breath biomarker for diagnosing lung cancer.” in the revision. (Page 3, Lines 56-58)

Q7: Line 52: This paragraph seems overly long and technical for an introduction. I would suggest reducing and putting some in the methods.

A7: Thank you for your suggestion. The introduction and methods section has been modified according to the Reviewer’s comments in the revision. (Pages 3-4, Lines 59-84, Pages 17-18, Lines 284-294)

Q8: Line 80: Poor grammar

A8: We are sorry for the mistake. We have corrected “In this study, the FENO levels of 740 breath gas samples (including 284 healthy control samples, and 456 primary lung cancer samples) were measured using a ringdown FENO analytic system to determine whether the FENO level was elevated in Ca, and the associations between different determinants (sex, age, smoking, etc.) and FENO were analyzed.” to “In this study, the exhaled NO levels of 740 breath gas samples (including 284 healthy control samples, and 456 primary lung cancer samples) were measured by ringdown exhaled NO analytic system. The system has the advantages of fast response, high data throughput, and validated accuracy. In addition to determine whether the exhaled NO level was elevated in LC, we also analyzed the associations between different determinants (sex, age, smoking, etc.) and exhaled NO.” in the revision. (Page 4, Lines 85-91)

Q9: Table 1: Were the lung cancer patients receiving any form of treatment or medication? This should be included.

A9: Thank you for your suggestion. We mentioned “A total of 740 breath gas samples were collected from 456 primary LC with pathologically diagnosed lung cancer without prior anti-cancer treatment (e.g., therapeutic agents, radiotherapy or chemotherapy)” in the patients section in the revision. (Page 5, Lines 97-99)

Q10: Line 111: Please add confidence interval values for AUC, sensitivity and specificity. I would also like to know (numerically) how many it got correct and what was the false positive rate. In addition, please provide the threshold for the ROC used.

A10: Thank you for your suggestion. we added confidence interval values for AUC, sensitivity, and specificity according to the Reviewer’s comments in the revision. The accuracy was 70.33%, the false positive rate was 0.45. ROC curve was drawn by Statistical Product and Service Solutions (IBM SPSS Statistics) version 25.0 (IBM Inc., USA). we used Youden Index to select sensitivity and specificity, and did not select the ROC threshold. Youden index = sensitivity + specificity − 1. (Pages 6-7, Lines 119-121)

Q11: Line 119: I found it odd that the methods nor the introduction discuss the use of electrochemical NO sensors – especially after the comments associated with them in the introduction. I think you should reduce this section and add it to the methods to say how the instrument was calibrated.

A11: Thank you for your suggestion. The results and methods section have been modified according to the Reviewer’s comments in the revision. (Page 17, Lines 295-302)

Q12: Line 143: You should be clear what the p value refers to. Be specific on the comparison.

A12: Thank you for your suggestion. We added the specific p value according to the Reviewer’s comments in the revision. (Pages 9, Line 155)

Q13: Line 177: Choosing 50 seems quite arbitrary. Would it not have been better to use the average age of a lung cancer sufferer?

A13: Thank you for your suggestion. we consider that the average age of lung cancer patients is 60, which is aging. The average age of healthy subjects is 47, which is younger, so choose the median age of 50.

Q14: Line 179: Should not start a sentence with “And”.

A14: Thank you for your suggestion. We have made changes in the revision. (Page 11, Line 191)

Q15: Lin 206: I would expect this to be linked with the breath analysis collection approach and I would like to see some comparison in the methods used.

A15: Thank you for your suggestion. We have added the methods used in other articles in the revision. (Pages 13-14, Lines 205-224).

Q16: Line 221: I would like to know the limitations of the study and further explanation for the results. For example, the analysis on cancer type is not discussed in the discussion. All the points raised in the results should be discussed/commented upon in this section. There is also no discussion of the clinical importance of the results. Is the sensitivity/specificity sufficient to be used as a screening test? What could be improved? These need to be included.

A16: Thank you for your suggestion. We added the limitations of the study and further explanation for the results according to the Reviewer’s comments in the revision. (Pages 14-15, Lines 238-247).

Q17: Line 250: Please add details of bag volume and air collected volume. It would be useful for the reader without the need to look at previous work.

A17: Thank you for your suggestion. The volume of FEP bags were 1 L (Page 16, Line 272), and We added the volume of breath sample provided by each subject was more than 0.7 L according to the Reviewer’s comments in the revision. (Page 16, Line 276)

Q18: Line 265: How was stability measured or is this from the manufacturer?

A18: We are very sorry for our negligence. In our previous work, reference 44, Figure 4. (a) shows the ringdown system’s stability tested in vacuum at the wavelength 226.255 nm. The baseline stability is defined as , where Δτ is the deviation and  is the average ringdown time. In the experiment, each data point of the baseline was obtained by averaging 300 ringdown events. The baseline stability at 226.255 nm is 0.52%.

Q19: Line 271: Brief details should be provided in the paper.

A19: Thank you for your suggestion. We provide brief details “This method is described in detail in our previous work, the exhaled NO level can be determined from a difference in the absorbance between a selected peak and the background baseline” according to the Reviewer’s comments in the revision. (Page 18, Lines 316-318)

We tried our best to improve the manuscript and made some changes in the manuscript. These changes will not influence the content and framework of the paper. And here we did not list the changes but marked in red in revised paper. Then we address major concerns below.

This study was designed as cases versus healthy control. Irrespective of their risk of biased results and limited applicability, these severe cases versus healthy-control designs have their place in test evaluation research. Indeed, without proof of concept, there is no need to further develop the tests. This study provided a p-value to indicate the difference between the levels of exhaled NO in the different groups, indicating a statistically significant difference between the cases and controls. However, at this stage, the sensitivity of patients with the disease and the specificity of healthy samples are low. The test performed poorly in such a design, so we can be quite sure the test will perform equally poorly in the clinical setting. Additional studies on the use of FE exhaled NO combined with other volatile organic compounds (VOCs) to develop a diagnosis model of lung cancer are needed.

About cost: the cost of the sample bags and disposable mouthpieces used in the experiment is very low. It costs about 1.5 million yuan to build a Ringdown exhaled NO analyzer. However, new detectors and beam sources with reduced sizes and costs are rapidly emerging.

Once again, thank you very much for your comments and suggestions.

Reviewer 2 Report

This is a good paper. The authors investigated exhaled nitric oxide (NO) for diagnosing lung cancer patients. They used an NO analyzer for breath analysis and tested 740 breath samples from 284 healthy control
subjects and 456 lung cancer patients. This is a very good number. They included medication, demographics, fasting status and smoking data. Lung Cancer patients had a significantly higher level of NO than the healthy referenced as a whole. But NO could not differentiate between the patients healthy people, the type of cancer, tumor node metastasis, sex, smoking status, age, and BMI.

It is a very good work and I support its publication 

1- It is important to also discuss possible effect of gut on exhaled NO that can interfere with lung NO (refer to Gastroenterology, 126, 2004 and Nat Rev Gastro & Hepat 16, 733, 2019). 

2- The use of abbreviation should be improved. For instance FENO is not correct as it reduces the readability of the paper. Instead exhaled NO is much better. Or usage of Ca instead of cancer patients should be altered 

3- Is there any reason that some patients produced more NO. Is there any information about the severity of cancer?

4- Include a final discussion about the usage of ingestible capsules in detecting the gut NO that can help in differentiating from the lung NO (ref to Nat Electron 1, 79, 2018)

5- There are a few small mistakes in references - like ref 43 that the year was repeated twice 

Author Response

Q1: It is important to also discuss possible effect of gut on exhaled NO that can interfere with lung NO (refer to Gastroenterology, 126, 2004 and Nat Rev Gastro & Hepat 16, 733, 2019).

A1: Thank you for your suggestion. The source of exhaled NO is reported, current thinking is that exhaled NO is formed in both the upper and lower respiratory tract and diffuses into the lumen by gaseous diffusion down a concentration gradient, thus conditioning exhaled gas with NO. There may be significant contribution from the oropharynx. Although gastric NO levels are very high, this does not appear to contaminate exhaled NO, probably because of closed upper and lower esophageal sphincters. (refer to ATS/ERS Recommendations for Standardized Procedures for the Online and Offline Measurement of Exhaled Lower Respiratory Nitric Oxide and Nasal Nitric Oxide, 2005)

Q2: The use of abbreviation should be improved. For instance FENO is not correct as it reduces the readability of the paper. Instead exhaled NO is much better. Or usage of Ca instead of cancer patients should be altered

A2: We are very sorry for our incorrect abbreviation. As Reviewer suggested that exhaled NO is used instead of FENO. And LC instead of lung cancer patients in the revision.

Q3: Is there any reason that some patients produced more NO. Is there any information about the severity of cancer?

A3: The level of exhaled NO has a strong correlation with T-helper cell 2 (Th2)-mediated airway inflammation and can help to diagnose this inflammation (refer to Bjermer L, Alving K, Diamant Z, et al. Current evidence and future research needs for exhaled NO measurement in respiratory diseases. Respir Med. 2014;108: 830–41). exhaled NO is closely related to the levels of nitric oxide synthase (NOS) in lung tissues. NOS can enhance the production of nitrite by alveolar macrophages in the lung tissues of patients with primary lung cancer (refer to American Thoracic S, European Respiratory S. ATS/ERS recommendations for standardized procedures for the online and offline measurement of exhaled lower respiratory nitric oxide and nasal nitric oxide, 2005. Am J Respir Crit Care Med. 2005;171: 912–30.). TNF-a, immunoglobulin E, interleukin (IL)4 and INF-y are regulatory factors for NO production (refer to Munoz-Fernandez MA, et, al. Activation of human macrophages for the killing of intracellular Trypanosoma cruzi by TNF-alpha and IFN-gamma through a nitric oxide-dependent mechanism. Immunol Lett. 1992;33: 35–40. and Kolb, et, al. Interleukin-4 stimulates cGMP production by IFN-gamma-activated human monocytes. Involvement of the nitric oxide synthase pathway. J Biol Chem. 1994;269: 9811–6.). In lung cancer, activated macrophages produce high NO levels that destroy or prevent the division of tumor cells by inhibiting DNA replication and preventing mitochondrial respiration (refer to American Thoracic S, European Respiratory S. ATS/ERS recommendations for standardized procedures for the online and offline measurement of exhaled lower respiratory nitric oxide and nasal nitric oxide, 2005. Am J Respir Crit Care Med. 2005;171: 912–30, and Moncada S, Palmer RM, Higgs EA. Nitric oxide: physiology, pathophysiology, and pharmacology. Pharmacol Rev. 1991; 43: 109–42.). NO may create a micro-environment that initiates tumorigenesis or promotes tumor heterogeneity, which can lead to metastasis (refer to Masri FA, et al. Abnormalities in nitric oxide and its derivatives in lung cancer. Am J Respir Crit Care Med. 2005;172: 597–605.).  All of these factors can result in higher mean exhaled NO levels in lung cancer patients.

According to the eighth edition TNM stage classification for lung cancer, the severity of lung cancer can be determined by staging information. In this study, the severity of lung cancer was not demonstrated by exhaled NO levels.

Q4: Include a final discussion about the usage of ingestible capsules in detecting the gut NO that can help in differentiating from the lung NO (ref to Nat Electron 1, 79, 2018)

A4: Thank you for your valuable comments. Next, we will study the metabolic mechanism of exhaled NO in lung cancer patients, and consider the use of ingestible capsules in detecting the gut NO that can help in differentiating from the lung NO.

Q5: There are a few small mistakes in references - like ref 43 that the year was repeated twice

A5: We are sorry for the mistake. We revised it in the revision. (Page 22, Line 483)

We tried our best to improve the manuscript and made some changes in the manuscript. These changes will not influence the content and framework of the paper. And here we did not list the changes but marked in red in revised paper. Once again, thank you very much for your comments and suggestions.

Reviewer 3 Report

This paper investigates the question as to whether exhaled nitric oxide levels could be used as a diagnostic tool to detect lung cancer. Here a reasonable number of patients and controls were sampled, with the conclusion that nitric oxide by itself would not be a useful tool for identifying lung cancer but could be used together with other tools to help with the diagnosis. This in itself is a very useful conclusion and could help in the development of suitable noninvasive methods. There are only small problems with the paper.  Is the ROC curve of Figure 2 affected by the skewed distribution of FENO and the disparity in between the number of controls versus number of cancer patients?

Author Response

Q: There are only small problems with the paper. Is the ROC curve of Figure 2 affected by the skewed distribution of FENO and the disparity in between the number of controls versus number of cancer patients?

A: Thanks for all your feedback and suggestions. ROC curves have an attractive property: they are insensitive to changes in class distribution. If the proportion of positive to negative instances changes in a test set, the ROC curves will not change. ROC graphs are based upon true positives rate and false positives rate, in which each dimension is a strict columnar ratio, so do not depend on class distributions (refer to Tom Fawcett, An introduction to ROC analysis, 27, 2006). We consider that the ROC curve of Figure 2 is not affected by the skewed distribution of exhaled NO and the disparity in between the number of controls versus number of cancer patients.

We tried our best to improve the manuscript and made some changes in the manuscript. These changes will not influence the content and framework of the paper. And here we did not list the changes but marked in red in revised paper. Once again, thank you very much for your comments and suggestions.

Reviewer 4 Report

Dear Authors, the manuscript is well written and well organised. I don't have specific comments on the text but I would like to understand the rationale behind the study. I strongly suggest to explain the relationship between FeNO and lung cancer, authors may include some points in the introduction section. This allows to understand the rationale of the study proposed by the authors. If this relationship is not clearly presented I have to say that the paper should be rejected.

Author Response

Q:I strongly suggest to explain the relationship between FeNO and lung cancer, authors may include some points in the introduction section. This allows to understand the rationale of the study proposed by the authors. If this relationship is not clearly presented I have to say that the paper should be rejected.

A: Thank you for your careful readings and valuable comments. The level of exhaled NO has a strong correlation with T-helper cell 2 (Th2)-mediated airway inflammation and can help to diagnose this inflammation (refer to Bjermer L, Alving K, Diamant Z, et al. Current evidence and future research needs for exhaled NO measurement in respiratory diseases. Respir Med. 2014;108: 830–41). exhaled NO is closely related to the levels of nitric oxide synthase (NOS) in lung tissues. NOS can enhance the production of nitrite by alveolar macrophages in the lung tissues of patients with primary lung cancer (refer to American Thoracic S, European Respiratory S. ATS/ERS recommendations for standardized procedures for the online and offline measurement of exhaled lower respiratory nitric oxide and nasal nitric oxide, 2005. Am J Respir Crit Care Med. 2005;171: 912–30.). TNF-a, immunoglobulin E, interleukin (IL)4 and INF-y are regulatory factors for NO production (refer to Munoz-Fernandez MA, et, al. Activation of human macrophages for the killing of intracellular Trypanosoma cruzi by TNF-alpha and IFN-gamma through a nitric oxide-dependent mechanism. Immunol Lett. 1992;33: 35–40. and Kolb, et, al. Interleukin-4 stimulates cGMP production by IFN-gamma-activated human monocytes. Involvement of the nitric oxide synthase pathway. J Biol Chem. 1994;269: 9811–6.). In lung cancer, activated macrophages produce high NO levels that destroy or prevent the division of tumor cells by inhibiting DNA replication and preventing mitochondrial respiration (refer to American Thoracic S, European Respiratory S. ATS/ERS recommendations for standardized procedures for the online and offline measurement of exhaled lower respiratory nitric oxide and nasal nitric oxide, 2005. Am J Respir Crit Care Med. 2005;171: 912–30, and Moncada S, Palmer RM, Higgs EA. Nitric oxide: physiology, pathophysiology, and pharmacology. Pharmacol Rev. 1991; 43: 109–42.). NO may create a micro-environment that initiates tumorigenesis or promotes tumor heterogeneity, which can lead to metastasis (refer to Masri FA, et al. Abnormalities in nitric oxide and its derivatives in lung cancer. Am J Respir Crit Care Med. 2005;172: 597–605.). All of these factors can result in higher mean exhaled NO levels in lung cancer patients. We added the relationship between exhaled NO and lung cancer according to the Reviewer’s comments in the revision. (Pages 3, Lines 48-49)

We tried our best to improve the manuscript and made some changes in the manuscript. These changes will not influence the content and framework of the paper. And here we did not list the changes but marked in red in revised paper. Once again, thank you very much for your comments and suggestions.

Round 2

Reviewer 4 Report

Dear Authors, thank you for explaining the relationship between NO and lung cancer. So that, I suggest to accept the manuscript in the actual version.